# Associations Between Sociocultural Attitudes Toward Appearance, Perfectionism, and Symptoms of Orthorexia Nervosa in Adolescent Football Athletes

**DOI:** 10.3390/healthcare13202625

**Published:** 2025-10-18

**Authors:** Wiktoria Staśkiewicz-Bartecka, Daniel Kandziora, Maksymilian Kafka, Paweł Marchewka, Agnieszka Białek-Dratwa, Agata Kiciak, Sylwia Jaruga-Sękowska, Daria Dobkowska-Szefer, Paweł Lewandowski, Samet Aktaş, Mateusz Grajek

**Affiliations:** 1Department of Food Technology and Quality Evaluation, Department of Dietetics, Faculty of Public Health in Bytom, Medical University of Silesia in Katowice, ul. Jordana 19, 41-808 Zabrze, Poland; s87425@365.sum.edu.pl (D.K.); s87383@365.sum.edu.pl (M.K.); akiciak@sum.edu.pl (A.K.); ddobkowska-szefer@sum.edu.pl (D.D.-S.); s83926@365.sum.edu.pl (P.L.); 2Faculty of Physical Culture Sciences, Jan Dlugosz University in Częstochowa, ul. Zbierskiego 6, 42-200 Czestochowa, Poland; pawelmarchewka1995@gmail.com; 3Department of Human Nutrition, Department of Dietetics, Faculty of Public Health in Bytom, Medical University of Silesia in Katowice, ul. Jordana 19, 41-808 Zabrze, Poland; abialek@sum.edu.pl; 4Department of Health Promotion, Faculty of Public Health in Bytom, Medical University of Silesia in Katowice, ul. Piekarska 18, 41-902 Bytom, Poland; sjaruga@sum.edu.pl; 5Department of Training Education, Faculty of Sports Science, Batman University, Yenişehir, 72000 Batman, Turkey; samet.aktas@batman.edu.tr; 6Department of Public Health, Faculty of Public Health in Bytom, Medical University of Silesia in Katowice, ul. Piekarska 18, 41-902 Bytom, Poland; mgrajek@sum.edu.pl

**Keywords:** adolescent athletes, football, orthorexia nervosa, perfectionism, sociocultural attitudes

## Abstract

**Background/Objectives:** Orthorexia nervosa and appearance-related pressures are increasingly discussed in youth sport, where performance demands may amplify perfectionistic tendencies and the internalization of cultural body ideals. This study examined how sociocultural attitudes toward appearance and perfectionism relate to orthorexic tendencies among adolescent football athletes. **Methods:** The study included players from a soccer school, with a final sample of 83 participants. All were Polish citizens aged 16–19. A cross-sectional design was used with standardized instruments: the Polish adaptation of the Düsseldorf Orthorexia Scale (DOS) to index symptoms of ON risk, the Sport Perfectionism Questionnaire (positive/negative perfectionism), and the SATAQ-3 subscales to assess sociocultural internalization/pressures and information exposure. **Results:** Across the entire sample (*n* = 73), most athletes were classified as having no risk of ON—60 people (82.2%), a smaller proportion showed an increased risk—10 people (13.7%), and symptoms of ON were found in 3 people (4.1%). In bivariate analyses, orthorexic tendencies co-occurred with perfectionism. In multivariate models, the addition of the perfectionism block provided a significant increase in explained variance over age, BMI, and sociocultural attitudes, while the SATAQ-3 block contributed only a small amount of additional variance in the presence of other predictors. **Conclusions:** Orthorexic risk is present but not widespread in adolescent football athletes. Perfectionistic tendencies emerge as salient psychosocial correlates of orthorexic symptoms, while sociocultural pressures appear relevant but partly overlapping and not uniquely predictive when modeled together.

## 1. Introduction

Symptoms of orthorexia nervosa (ON) constitute one of the most recent challenges in the field of health psychology and research on eating disorders. The term was first introduced by Steven Bratman in 1998 and was based on clinical observations as well as the author’s personal experiences in which he described the obsessive need to control the quality of consumed food [1,2]. Since then, the concept has attracted considerable attention from researchers, and numerous studies indicate that symptoms of orthorexia nervosa are associated with a pathological focus on the quality of food, its origin, processing method, or even the type of packaging [3,4,5]. A characteristic feature of this disorder is the presence of persistent thoughts related to eating, which may sometimes lead to social isolation, nutritional deficiencies, or deterioration of psychological well-being [6,7,8]. Despite the growing number of studies, there is still no universally accepted definition or standardized diagnostic criteria, which limits the comparability of results and hinders the development of coherent preventive and therapeutic strategies [6,9].

Symptoms of ON often co-occur with other mental health problems such as obsessive compulsive disorder, depression, or other forms of eating disorders (EDs) [4,8]. Groups particularly at risk include individuals with perfectionistic traits, students of medical and health-related fields as well as athletes [9,10]. In the context of sport, especially competitive sport, the relationship between perfectionism and ON appears to be particularly significant. Perfectionism, understood within the two-dimensional model as the distinction between adaptive perfectionism (strivings; positive) and maladaptive perfectionism (concerns; negative), plays a key role in the functioning of young athletes [11,12]. Numerous studies indicate that perfectionistic strivings may foster engagement and performance improvement, whereas perfectionistic concerns increase vulnerability to anxiety, burnout, and EDs [13,14,15,16].

Young football players represent a particularly relevant research group as they are in the period of adolescence, during which identity is formed while social and athletic pressures increase. Football, as the most globally popular sport, combines high physical demands with considerable social exposure, making players vulnerable to social comparisons and the internalization of cultural appearance standards [17,18]. Studies employing the SATAQ-3 (Sociocultural Attitudes toward Appearance Questionnaire) have demonstrated that the internalization of beauty ideals promoted by the media is a significant predictor of body dissatisfaction and EDs [19]. Combined with coaching pressure, family expectations, and the influence of social media, these factors may foster the development of orthorexic and perfectionistic behaviors among young football players [20,21].

Previous research on symptoms of orthorexia nervosa in sport has primarily focused on adult athletes or students in health-related fields [22,23]. However, there is a lack of studies addressing adolescents training in professional football academies, a group particularly vulnerable to perfectionism-related pressure, both in terms of athletic performance and physical appearance. This gap is significant, as adolescence (16–19 years) represents a critical period for psychological development, body image formation, and susceptibility to EDs [24,25]. Understanding the relationships between ON, perfectionism, and the internalization of sociocultural ideals may provide not only theoretical insights, but also practical implications for the development of preventive and intervention programs within the context of sports academies.

The aim of the present study was to assess the relationships between ON, sport-related perfectionism, and the internalization of appearance-related attitudes among adolescents aged 16–19 years training at a football academy. It was hypothesized that higher levels of maladaptive perfectionism and stronger internalization of cultural norms would be positively correlated with the severity of orthorexic behaviors. Additionally, it was assumed that adaptive perfectionism may serve a protective function by reducing the risk of developing maladaptive attitudes toward eating and body image. The corresponding null hypotheses were as follows: H0_1_: maladaptive perfectionism is not associated with the severity of orthorexic behaviors. H0_2_: internalization of cultural norms is not associated with the severity of orthorexic behaviors. H0_3_: adaptive perfectionism is not associated with a reduction in maladaptive eating and body-image attitudes.

## 2. Materials and Methods

### 2.1. Procedure of the Study

The study was conducted in March 2025 and included a population of players from a football academy located in the Silesian Metropolis area. The data collection process lasted seven days, which ensured a high degree of uniformity of conditions and comparability of the results obtained across all participants. Self-administered questionnaires were distributed and completed in the school environment immediately after anthropometric measurements were taken. To reduce the risk of incorrect responses and ensure full comprehension of the questionnaire content, all participants received detailed instructions on how to complete the surveys beforehand.

Prior to participation, all players—and, in the case of minors, also their legal guardians—were thoroughly informed about the study’s aims and procedures, the principles of voluntary participation, data processing methods, and the guarantees of anonymity and confidentiality of responses. Informed consent was obtained from each participant (and their legal guardian, if applicable) before the commencement of the research process.

The study was carried out in accordance with the ethical guidelines of the World Medical Association’s Declaration of Helsinki. Ethical approval for the research was granted by the Bioethics Committee of the Medical University of Silesia in Katowice (No. BNW/NWN/0043-3/641/35/23) on 22 September 2023.

### 2.2. Participants

A non-probabilistic (total population sampling) approach was used: all eligible football players from a single Polish football academy were invited (frame N = 85). After applying pre-specified exclusion criteria (e.g., lack of consent, incomplete data), the final analytic sample was *n* = 73; throughout the manuscript, *n* refers to this post-exclusion sample. All participants were of Polish nationality, aged 16–19 years (M = 17.26; SD = 1.09). The upper age limit of 19 years was adopted to capture developmental and psychosocial processes typical of late adolescence, a period characterized by intensifying training demands, increasing performance pressure, and heightened vulnerability to body-image–related difficulties.

The inclusion criteria were as follows: (1) obtaining informed consent from the participant and/or their legal guardian, (2) age between 16 and 19 years, (3) active participation in academy and training activities at the time of the study, and (4) no injuries resulting in a break from training lasting ≥7 consecutive days within the two months preceding the study. The exclusion criteria included: (1) incomplete or incorrectly completed questionnaires, and (2) absence during the data collection session.

### 2.3. Survey Tools

A multisectional questionnaire was employed in the study, comprising both original research questions and standardized psychometric instruments. The introductory section was designed to collect sociodemographic and anthropometric data including age, height, body mass, presence of chronic diseases, and long-term medication use. Additional information was gathered regarding physical activity undertaken outside of football school training sessions, dietary practices involving the elimination of specific food products, and the use of social media.

Subsequent sections incorporated validated psychometric instruments: the Perfectionism in Sport Questionnaire (PSQ) [26], the Düsseldorf Orthorexia Scale (DOS) [27], and the Sociocultural Attitudes toward Appearance Questionnaire—third version (SATAQ-3) [28].

#### 2.3.1. BMI-Based Weight Status (Screening Indicator)

Anthropometric measurements, including body weight and height, were taken prior to the questionnaire administration. Body height was recorded using a SECA 756 medical scale (Seca GmbH & Co. KG., Hamburg, Germany). Based on the obtained values, the body mass index (BMI) was calculated by dividing the body weight (kg) by the square of body height (m^2^).

BMI values were further analyzed in comparison with age- and sex-specific percentile grids developed for the Polish population as well as for adults according to the WHO guidelines [29,30]. BMI-based weight status classification for individuals under 18 years of age followed the national standards: BMI values ≤ 5th percentile were interpreted as underweight, values between the 5th and 85th percentile indicated normal weight, values from ≥85th to <95th percentile indicated overweight, and values ≥95th percentile were classified as obesity [30].

#### 2.3.2. Perfectionism in Sport Questionnaire (PSQ)

Perfectionism was assessed using the Polish version of the psychometric instrument PSQ. It consists of 30 items, of which 13 form the positive perfectionism (PP) dimension and 17 correspond to the negative perfectionism (NP) dimension. Responses were provided on a five-point Likert scale, where 1 indicated “strongly disagree” and 5 indicated “strongly agree” [26].

A semantic analysis of the item content allowed for the identification of the key components of each dimension. The PP scale reflected, among others, high personal standards, focus on skill development, and satisfaction and enjoyment derived from sports participation. In contrast, the NP scale was associated with fear of making mistakes, excessive focus on failures, intensified emotional reactions to defeats, and discouragement resulting from discrepancies between intended goals and actual achievements [26]. The questionnaire was developed and validated by the authors of the scale; the reliability of the tool was determined using Cronbach’s alpha, which reached 0.91 for the PP scale and 0.93 for the NP scale, indicating a high level of validity and reliability for both dimensions [31].

The reliability of both subscales was additionally estimated using McDonald’s omega coefficient. The obtained values—0.82 for PP and 0.82 for NP—indicate a high level of internal consistency of the instrument.

#### 2.3.3. Düsseldorf Orthorexia Scale (DOS)

The first tool developed for the assessment of ON was the Bratman test, published in 2000, followed by the ORTO-15 questionnaire created by Donini and colleagues [2]. Despite the widespread use of ORTO-15 in research, this tool has faced criticism due to questionable validity and inconsistent cutoff points [7,10]. Therefore, in recent years, the Düsseldorf Orthorexia Scale (DOS) has gained increasing popularity. The questionnaire has already been translated and validated in several languages, including Polish, which significantly enhances its comparative and practical value [4].

The DOS is a screening tool developed for the assessment of eating behaviors characteristic of ON [27]. The shortened 10-item version of the scale, treated as a unidimensional measure of orthorexic tendencies, was derived from the full 21-item questionnaire, which originally included subscales related to orthorexic behaviors, avoidance of food additives, and mineral consumption [32].

In the present study, the shortened version of the DOS was applied. Participants responded to 10 statements using a four-point Likert scale, ranging from “strongly disagree” to “strongly agree”. The scale did not include reverse-scored items. The maximum possible score was 40 points, with interpretation consistent with the authors’ recommendations: scores >30 indicated the presence symptoms of ON, scores of 25–29 suggested an elevated risk, whereas scores <25 did not indicate the presence of the disorder [27].

The study utilized the Polish adaptation of the scale (PL-DOS), developed and validated by the authors of the original version, which demonstrated reliability comparable to the E-DOS scale; Cronbach’s α reached 0.84 [27,32]. The reliability analysis of PL-DOS results, conducted using McDonald’s ω coefficient, yielded a value of 0.816, confirming an acceptable level of internal consistency of the tool.

#### 2.3.4. Sociocultural Attitudes Toward Appearance Questionnaire—Wersja Trzecia (SATAQ-3)

The SATAQ-3 is one of the most widely used instruments for assessing the influence of sociocultural norms, promoted by mass media, on attitudes and behaviors related to the body and physical appearance. Developed by Heinberg and Thompson [28], this questionnaire—similar to its earlier versions—enables the evaluation of the degree of pressure and internalization of standards associated with body image and physical attractiveness [33]. In the present study, the Polish adaptation of the questionnaire developed by Izydorczyk and Lizińczyk [34] was used.

The Polish adaptation of the SATAQ-3 consists of 28 items and includes four factors: Internalization-Pressure (IP; 12 items), Internalization-Information Seeking (I-IS; 6 items), Internalization-Sport (IS; 4 items), and Information (I; 6 items). Cronbach’s α values for the subscales ranged from 0.76 to 0.92, confirming satisfactory reliability in studied populations of women and men in Poland [34].

The overall internal consistency of the scale, estimated using McDonald’s ω coefficient, was 0.914. For the subscales, the following values were obtained: Internalization-Pressure (ω = 0.927), Internalization-Sport (ω = 0.854), Internalization-Information Seeking (ω = 0.816), and Information (ω = 0.830).

### 2.4. Statistical Analysis

Statistical analyses were performed using Statistica version 13.3 (StatSoft, Kraków, Poland) and R software (version 4.0.0; R Foundation for Statistical Computing, Vienna, Austria, 2020), operating under the GNU General Public License (GPL). Quantitative data were summarized using descriptive statistics including means (X), standard error (SE), median (Med), standard deviations (SD), minimum (min), maximum (max), skewness (skew), and skewness error (skewE). Qualitative data were expressed as percentages. The normality of the data distribution was assessed using the Shapiro–Wilk test.

Differences between categorical variables (e.g., response distributions across age groups 16–19 years) were tested with the chi-square test of independence; Fisher’s exact test or corrections were applied when the expected frequencies were low.

Relationships between quantitative variables were examined using Pearson’s correlation, reporting the correlation coefficient (r), degrees of freedom, and *p*-values.

The association between symptoms of ON severity (DOS total score) and potential predictors was assessed with hierarchical multiple linear regression: Model 1 (age, BMI), Model 2 (SATAQ-3 subscales), and Model 3 (perfectionism: PP, NP). The analyses reported b (SE), 95% CI, *t*, R^2^, and ΔR^2^ for model fit increments.

The primary dependent variable was the severity of orthorexic tendencies, operationalized as the DOS total score. Independent variables were psychosocial predictors measured as continuous totals: perfectionism (PP and NP) and sociocultural appearance pressures (SATAQ-3 subscales: Internalization-Pressure [IP], Internalization-Athlete [IA], Internalization-Information Seeking [I-IS], and Information [I]). Age (years) and BMI (kg/m^2^) were included as covariates in all regression models.

The internal reliability of the scales (PSQ, DOS, SATAQ-3 subscales) was estimated using McDonald’s omega (ω).

The level of statistical significance was set at *p* < 0.05.

## 3. Results

Table 1 presents the characteristics of the study participants including age, height, body mass, and BMI. The mean age of the participants was 17.26 ± 1.09 years (range: 16–19). The mean body height was 180.23 ± 6.08 cm, and body mass ranged from 50 to 93 kg, with a mean of 69.74 ± 8.17 kg. According to the adopted interpretation of BMI values, one player was classified as underweight, while three players were classified as overweight. The remaining participants had normative body mass.

Additional physical activity outside the football club was declared by 65 players. The most frequently reported frequency was 3–4 times per week (*n* = 29) or 1–2 times per week (*n* = 24). No significant associations were found between age and undertaking additional physical activity (*p* = 0.10), nor between age and its frequency (*p* = 0.17).

In the total sample (*n* = 73), 10 participants (13.7%) were classified as being at elevated risk of ON, and 3 (4.1%) were diagnosed with symptoms of ON; the remaining 60 (82.2%) were classified as not at risk. No significant differences were found between age groups (*p* = 0.321).

Analysis of social media activity showed that most participants spent 2 to 3 h daily on these platforms (27.4% and 37.0%, respectively) regardless of the risk of ON (*p* = 0.901). The most frequently chosen platforms were Instagram (35.6%) and TikTok (34.2%), followed by Snapchat (11.0%). The main declared purpose of using social media was entertainment and relaxation (58.9%), followed by maintaining contact with friends (21.9%). Posting content and following influencers/topics were reported only occasionally (5.5% each).

With regard to body satisfaction, the majority of participants described themselves as satisfied but willing to make some changes (75.3%). Only 19.2% of participants were completely satisfied with their appearance, whereas 5.5% reported dissatisfaction. Differences between groups according to DOS interpretation were not statistically significant (*p* = 0.179). Detailed information is presented in Table 2.

Significant positive correlation were observed between perfectionism and indicators of orthorexic tendencies: DOS correlated with PP (r = 0.312; *p* = 0.007) and with NP (r = 0.283; *p* = 0.015). The strongest relationship within perfectionism traits was noted between PP and NP (r = 0.760; *p* < 0.001).

In the domain of sociocultural attitudes toward appearance, the Internalization of the Athletic Ideal scale was moderately positively correlated with PP (r = 0.297; *p* = 0.011) and NP (r = 0.268; *p* = 0.022) as well as with Internalization-Pressure (r = 0.450; *p* < 0.001) and Information (r = 0.335; *p* = 0.004). At the same time, it showed a significant negative correlation with the Internalization-Information Seeking scale (r = −0.242; *p* = 0.039). The Internalization-Pressure and Information scales were very strongly interrelated (r = 0.873; *p* < 0.001).

BMI did not show significant correlation with DOS, PP, NP, or the SATAQ-3 subscales (all *p* > 0.05).

The findings suggest that higher levels of perfectionism (particularly its negative component) co-occur with greater severity of orthorexic tendencies, and that the internalization of the athletic body ideal is consistent with greater pressure and informational exposure to appearance-related content. At the same time, BMI remained independent of the psychological constructs examined (Table 3).

Table 4 presents the results of the hierarchical regression for symptoms of ON severity (DOS scores). Model 1 (age, BMI) explained R^2^ = 0.026. The addition of the SATAQ-3 subscales (Model 2) did not significantly improve the model fit (*p* = 0.903). The inclusion of perfectionism (PP, NP) in Model 3 significantly increased the explained variance to R^2^ = 0.174 (*p* = 0.008). In the final model, however, none of the individual predictors (age, BMI, SATAQ-3 subscales, PP, NP) reached statistical significance (all *p* > 0.05).

## 4. Discussion

The research problem addressed in this study is of considerable importance from the perspective of sport psychology, dietetics, and the prevention of EDs. During late adolescence, football academy players face increasing demands regarding athletic performance as well as sociocultural pressures related to appearance. These factors may contribute to the development of orthorexic behaviors and the intensification of perfectionism, which in turn may lead to a deterioration in the psychological and physical well-being of young athletes. Our findings align with this area, providing data on the proportion of participants at risk of ON and on the relationships between perfectionism, the internalization of appearance ideals, and health-related behaviors in a group of young football players.

The analysis of the results showed that most participants did not present signs of ON, although 13.7% were found to be at elevated risk and 4.1% were identified as displaying orthorexic behaviors. These findings indicate that the problem is not widespread; however, the presence of a minority group of players at elevated risk of orthorexic symptoms warrants diagnostic and preventive attention. Importantly, significant positive correlations were observed between the severity of orthorexic tendencies and perfectionism—both positive (*p* = 0.007) and negative (*p* = 0.015). The internalization of the athletic body ideal was significantly associated with perfectionism components and exposure to appearance-related content (e.g., IA—IP: *p* < 0.001; IA—I: *p* = 0.004; IA—I-IS: *p* = 0.039), suggesting that sociocultural pressure plays a reinforcing role in shaping orthorexic attitudes. Among the tested blocks, adding perfectionism (PP, NP) was associated with the largest improvement in model fit, although none of the individual coefficients reached statistical significance.

The prevalence and characteristic symptoms of ON in sports, particularly among young football players, are increasingly addressed in the scientific literature. The results of our study indicate that although the proportion of players meeting the criteria for elevated ON risk was not high, the presence of this group signals a potential problem that may intensify with increasing training and social pressure. Similar observations were made by Zydek et al. [20], who, when analyzing young football players, demonstrated significant associations between perfectionism and orthorexic tendencies, indicating that this factor largely determines the risk of disorders regardless of objective body composition indicators [20].

In the context of sex differences, Staśkiewicz-Bartecka et al. [21] showed that female football players, particularly those influenced by social media and the internalization of body ideals, presented a higher risk of ON, underscoring the importance of cultural and media pressures as factors exacerbating behaviors associated with obsessive dietary control [21]. Furthermore, studies comparing amateur and professional players revealed that athletes at higher competitive levels more frequently displayed attitudes indicative of symptoms of ON risk, which may be explained by the stronger pressure to achieve results and maintain an “ideal” physique [22]. In light of our findings—where only a small proportion of participants met the DOS-based screening criteria for elevated risk of orthorexic symptoms—it is important to note that risk correlates may differ by age, gender, and level of athletic involvement. Overall, the pattern is consistent with a multifactorial account of ON in youth sport, reflecting multiple influences (individual predispositions and environmental/contextual exposures); however, these results indicate associations, not causation.

Perfectionism revealed in the present findings played a key role, which is consistent with previous research in sports. The two-dimensional model of perfectionism highlights both adaptive and maladaptive aspects of this construct [35]. While positive perfectionism is associated with commitment and the desire to improve skills, negative perfectionism constitutes a risk factor for burnout, anxiety, and eating disorders [36]. In this study, both PP and NP correlated with the severity symptoms of ON, which may indicate that even seemingly adaptive striving for excellence in the context of youth sports can lead to excessive control over eating behaviors.

Research by Smith et al. [37] demonstrated that an appropriate motivational climate created by coaches can reduce sports anxiety and mitigate the negative effects of perfectionistic expectations [37]. Similarly, Jordana et al. [38] showed that among academy football players, negative perfectionism often co-occurs with irrational beliefs and deteriorating mental health [38]. The long-term consequences of negative perfectionism were documented by Smith et al., who indicated its association with burnout and depression in young football players, suggesting that early symptoms may manifest in lowered body image self-esteem and later progress into more serious emotional difficulties [14]. On the other hand, studies by Larkin et al. [39] emphasize that high levels of perfectionism may enhance training engagement and determination, albeit at the cost of greater susceptibility to pressure and stress [39]. Similar conclusions were drawn by Hill et al. [40], where a lack of unconditional self-acceptance mediated the relationship between perfectionism and burnout in young athletes [40].

Taken together with our observations, these findings suggest that early identification of negative perfectionism, alongside the reinforcement of self-compassion and realistic training goals, may serve a protective function, minimizing the risk of developing mental health problems and a decline in athletic performance.

The third area of interpretation concerns social media and sociocultural pressures. In the present study, players most frequently reported using Instagram and TikTok, with the main purpose being entertainment and maintaining contact with friends. Although no significant differences between groups were observed in relation to DOS, the correlation results indicated an association between the internalization of the athletic body ideal, perfectionism, and informational exposure. This confirms the conclusions of Burgon et al. [18], who pointed out that social media foster social comparisons and increase the risk of body dissatisfaction among athletes [18]. Similarly, Babic et al. [17] emphasized that exposure to media messages undermines adolescents’ positive physical self-esteem [17]. Our findings suggest that while time spent on social media did not differentiate ON risk, the internalization of aesthetic norms remains an important psychological correlate.

The practical implications of these findings are particularly relevant in the context of working with young athletes in football academies. The results highlight the need for systematic monitoring of both nutritional attitudes and personality traits, such as perfectionism, which may contribute to the development of eating disorders. It appears crucial to implement preventive programs that integrate nutritional education with mental resilience training, self-compassion, and realistic goal setting. It is recommended that coaches and sports psychologists receive training in recognizing early warning signs and symptoms of ON and negative perfectionism and in creating a motivational climate that promotes balance between athletic development and psychological well-being. Furthermore, given the role of social media, it is essential to cultivate a critical stance toward media messages and to promote skills for the conscious and mindful use of these platforms. Such integrated actions may not only minimize the risk of developing eating disorders, but also support the long-term psychophysical development of young football players and their ability to cope with the pressures of athletic and social demands.

One of the key strengths of the present study is its innovative character, consisting of the simultaneous analysis of three areas. This approach allowed us to obtain a comprehensive picture of the psychological and environmental factors that may influence health-related behaviors during late adolescence. Another strength is the homogeneity of the group in terms of age, training status, and athletic level, which increases the internal consistency of the study, limits the impact of confounding factors, and enables a more precise interpretation of the results. The use of standardized, reliable psychometric instruments such as DOS, SATAQ-3, and PSQ represents an additional advantage that enhances the credibility and comparability of the findings with other studies. An important asset is also the control of research conditions—the measurements were conducted in the same academy and at similar times of day. This approach minimized the risk of response bias and increased participant comfort. Despite these strengths, the study also has limitations. The most important is the use of a cross-sectional design. Another limitation is that the study included only young male football players from a single sports academy. Therefore, the results cannot be directly generalized to female players, other sports disciplines, or international populations. A further limitation is the potential bias resulting from the use of self-report instruments. Additionally, some important contextual variables, such as socioeconomic status or the level of family support, were not fully considered, which may have influenced the interpretation of the findings.

## 5. Conclusions

In the studied population of young football players, most participants did not exhibit orthorexic symptoms; however, a minority group at elevated risk and isolated cases with confirmed ON symptoms were identified. With respect to the hypotheses: higher maladaptive perfectionism correlated with greater ON symptom severity in bivariate analyses but did not remain an independent predictor after adjustment; internalization of appearance norms did not show an independent association with ON symptoms in multivariable models; a protective effect of adaptive perfectionism was not supported. The findings are associational and do not permit causal interpretation. From a healthcare perspective aimed at strengthening the mental well-being of adolescent footballers, we recommend routine screening for ON symptoms, perfectionistic concerns, and body-image difficulties; brief evidence-informed interventions targeting components of perfectionism, media literacy and the critical appraisal of appearance-related content, and self-compassion; collaboration with coaches to foster a task-involving climate and limit weight-centric messaging; and clear referral pathways to sport-psychology/clinical services for at-risk athletes.

Future research should employ prospective cohort, time-ordered designs in larger and more diverse samples and use multimethod assessment to more precisely determine the direction of associations and the mechanisms linking internalization, perfectionism, and orthorexic symptoms.

## Figures and Tables

**Table 1 healthcare-13-02625-t001:** Characteristics of the group (*n* = 73).

	M	SE	Med	SD	Min	Max	Skew	SkewE
Age (years)	17.3	0.1	17.0	1.1	16.0	19.0	0.3	0.3
Height [cm]	180.2	0.7	181.0	6.1	159.0	190.0	−0.8	0.3
Body mass [kg]	69.7	1.0	69.0	8.2	50.0	93.0	0.4	0.3
BMI [kg/m^2^]	21.4	0.2	21.1	2.0	17.1	27.8	0.6	0.3

M—mean; SE—error standard; Med—median; SD—standard deviation; Min—minimum; Max—maximum; Skew—skewness; SkewE—standard skew error.

**Table 2 healthcare-13-02625-t002:** Social media use in relation to the risk of ON: daily usage time, most frequently used platforms, purposes of use, and body satisfaction (*n* = 73).

		DOS Interpretation			
Questions	Responses	Presence ON (*n* = 3)	Risk of ON (*n* = 10)	No Risk (*n* = 60)	TOTAL (*n* = 73)
Time (How much time per day do you spend on social media?)	1 h	0 (0.0%)	0 (0.0%)	2 (3.3%)	2 (2.7%)
	2 h	1 (33.3%)	4 (40.0%)	15 (25.0%)	20 (27.4%)
	3 h	2 (66.7%)	3 (30.0%)	22 (36.7%)	27 (37.0%)
	4 h	0 (0.0%)	2 (20.0%)	10 (16.7%)	12 (16.4%)
	More than 4 h	0 (0.0%)	1 (10.0%)	11 (18.3%)	12 (16.4%)
	*p*-value	0.901			
Platform (Which social media platforms do you use most frequently?)	Discord	0 (0.0%)	0 (0.0%)	1 (1.7%)	1 (1.4%)
	Facebook	0 (0.0%)	1 (10.0%)	3 (5.0%)	4 (5.5%)
	Instagram	0 (0.0%)	7 (70.0%)	19 (31.7%)	26 (35.6%)
	Snapchat	1 (33.3%)	0 (0.0%)	7 (11.7%)	8 (11.0%)
	TikTok	1 (33.3%)	2 (20.0%)	23 (38.4%)	26 (35.6%)
	Twitch	0 (0.0%)	0 (0.0%)	2 (3.3%)	2 (2.7%)
	X (Twitter)	0 (0.0%)	0 (0.0%)	4 (6.7%)	4 (5.5%)
	YouTube	1 (33.3%)	0 (0.0%)	1 (1.7%)	2 (2.7%)
	*p*-value	0.182			
Purpose (For what purpose do you primarily use social media?)	Sharing experiences (publications, achievements, important events)	0 (0.0%)	1 (10.0%)	5 (8.3%)	6 (8.2%)
	Contact with friends	0 (0.0%)	3 (30.0%)	13 (21.7%)	16 (21.9%)
	Learning	0 (0.0%)	1 (10.0%)	3 (5.0%)	4 (5.5%)
	Following influencers/celebrities	0 (0.0%)	1 (10.0%)	3 (5.0%)	4 (5.5%)
	Entertainment/relaxation	3 (100%)	4 (40.0%)	36 (60.0%)	43 (58.9%)
	*p*-value	0.872			
Satisfaction with appearance (Are you satisfied with your physical appearance?)	I am satisfied and would not change anything	1 (33.3%)	2 (20.0%)	11 (18.3%)	14 (19.2%)
	I am satisfied but would change a few things	1 (33.3%)	7 (70.0%)	47 (78.3%)	55 (75.3%)
	I am not satisfied	1 (33.3%)	1 (10.0%)	2 (3.3%)	4 (5.5%)
	*p*-value	0.179			

**Table 3 healthcare-13-02625-t003:** Correlation matrix between BMI, DOS, PP, NP, SATAQ-3 subscales, and age (*n* = 73).

		BMI	DOS	PP	NP	IP	IA	I-IS	I	Age
BMI	r	—								
	df	—								
	*p*	—								
DOS	r	−0.01	—							
	df	71	—							
	*p*	0.948	—							
PP	r	0.06	0.31	—						
	df	71	71	—						
	*p*	0.635	0.007 **	—						
NP	r	0.09	0.28	0.76	—					
	df	71	71	71	—					
	*p*	0.430	0.015 *	< 0.001 ***	—					
IP	r	−0.02	0.05	−0.03	0.05	—				
	df	71	71	71	71	—				
	*p*	0.893	0.648	0.799	0.698	—				
IA	r	0.06	0.02	0.30	0.27	0.45	—			
	df	71	71	71	71	71	—			
	*p*	0.618	0.887	0.011 *	0.022 *	< 0.001 ***	—			
I-IS	r	−0.04	−0.05	0.01	−0.08	−0.21	−0.24	—		
	df	71	71	71	71	71	71	—		
	*p*	0.765	0.666	0.905	0.513	0.079	0.039 *	—		
I	r	−0.04	0.12	−0.04	−0.05	0.87	0.34	−0.22	—	
	df	71	71	71	71	71	71	71	—	
	*p*	0.712	0.323	0.735	0.699	< 0.001 ***	0.004 **	0.061	—	
Age	r	0.39	−0.15	0.03	0.14	−0.16	−0.14	0.13	−0.23	—
	df	71	71	71	71	71	71	71	71	—
	*p*	< 0.001 ***	0.205	0.823	0.241	0.174	0.224	0.273	0.051	—

BMI—body mass index; DOS—Düsseldorf Orthorexia Scale; PP—positive perfectionism; NP—negative perfectionism; IP—Internalization-Pressure subscale; IA—Internalization-Athlete; I-IS—Internalization-Information Seeking; I—Information; r—Pearson’s correlation coefficient; df—degrees of freedom; *p*—significance level; * = *p* < 0,05; ** = *p* < 0,01; *** = *p* < 0.001.

**Table 4 healthcare-13-02625-t004:** Hierarchical regression for DOS: model fit indices, model comparisons, and coefficients of the final model.

**Model Fit Indices**						
**Model**	**R**	**R^2^**				
1	0.160	0.026				
2	0.20	0.04				
3	0.42	0.17				
**Model Comparisons**						
**Model**	**ΔR^2^**	**F**	**df1**	**df2**	** *p* **	
1–2	0.02	0.26	4	66	0.903	
2–3	0.13	5.15	2	64	0.008	
**Model Coefficients-DOS**						
**Predictor**	**Estimate**	**SE**	**95% CI**		** *t* **	** *p* **
			**Lower**	**Upper**		
Intercept	17.9	11.1	−4.3	40.0	1.6	0.112
BMI	0.1	0.3	−0.5	0.7	0.4	0.683
Age	−0.8	0.6	−2.0	0.4	−1.4	0.162
IP	−0.1	0.1	−0.3	0.2	−0.6	0.568
IA	−0.2	0.1	−0.5	0.1	−1.1	0.259
I-IS	−0.0	0.1	−0.2	0.2	−0.2	0.854
I	0.3	0.3	−0.2	0.8	1.1	0.272
PP	0.2	0.2	−0.1	0.5	1.2	0.253
NP	0.1	0.1	−0.1	0.3	1.1	0.286

BMI—body mass index; DOS—Düsseldorf Orthorexia Scale; Model 1: age, BMI; Model 2: subscales SATAQ-3 (IP—Internalization-Pressure subscale; IA—Internalization-Athlete; I-IS—Internalization-Information Seeking; I—Information); Model 3: PSQ (PP—positive perfectionism; NP—negative perfectionism); R—multiple correlation; R^2^—coefficient of determination; ΔR^2^—change in R^2^ relative to the previous model; F—F statistic for the ΔR^2^ test (incremental fit test); df1/df2—degrees of freedom for the F test; SE—standard error; 95% CI—95% confidence interval; *t*—*t* statistic; *p*—significance level; Intercept—constant term.

## Data Availability

The raw data supporting the conclusions of this article will be made available by the authors on request.

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
