# Peer review of "Associations Between Sociocultural Attitudes Toward Appearance, Perfectionism, and Symptoms of Orthorexia Nervosa in Adolescent Football Athletes"

_healthcare, 2025, doi:10.3390/healthcare13202625_

Round 1
Reviewer 1 Report
Comments and Suggestions for Authors
I have reviewed the manuscript titled 'Associations Between Sociocultural Attitudes Toward Appearance, Perfectionism, and Orthorexia Nervosa in Adolescent Football Athletes'.
As a consequence, I have serious concerns as follows:
- Suggestion: In the manuscript, the term 'orthorexia' could be changed to 'symptoms of orthorexia'.
- I recommend that the Authors should return keywords in alphabetical order.
- Lines 58-65: This information is more methodological oriented. I recommend that the Authors could move this paragraph to the section of Materials and Methods.
- Lines 101-105: The Authors put forward a number of alternative hypotheses. Also, I recommend that the Authors could write down null hypotheses.
- 2.2. Participants: How was the representative sample size calculated? How large was the marginal error applied?
- 2.2. Participants: What was the technique used to select the study participants: probabilistic or non-probabilistic?
- 2.2. Participants: I propose that the number of participants (n) in the survey be adjusted in the light of the exclusion criteria.
- Lines 150-153: Add the references, please.
- Nutritional Status: The Authors should be aware that the body mass index is not a nutritional status. Additional research on the intake of macro-nutrients and eating habits appear necessary. Therefore, I suggest the Authors to decline writing about the nutritional status.
- Line 228: 'Associations between...' should be changed to 'Differences between...'.
- Line 233: . '..predictors was assessed... ' should be changed '...potential predictors...'. The Authors should be aware that they have carried out a single cross-sectional study in design and that the use of the causal language must be minimized throughout the entire manuscript.
- In the section, namely, Materials and Methods, must be indicated variables, which have been assigned to independent variables and dependent ones.
- Results: I recommend that all numbers (except p-value) should be rounded to one decimal place.
- Table 1: 'Age [years]' must changed to 'Age (years)'.
- Table 1: 'X—average'. This symbol 'X' is not the mean. This information needs to be corrected.
- Lines 250-257: In the Reviewer’s view, the information on those lines is irrelevant.
- Line 260: '…significant associations...'. The Authors should be aware that the identification of an association is a subject obtained from regression analysis. Otherwise, the concept 'correlation' should be used.
- Line 267: 'The most frequently chosen platforms were Instagram (35.6%) and TikTok (34.2%), followed by Snapchat (11.0%)…'. Can the Authors explain why this information is important? I suggest that the Authors could write the Results section in a more consistent way and give only the most important results.
- Line 279: 'Significant positive associations were observed…' must be changed to '…correlation…'.
- Line 305-307: 'Model 3 significantly increased the explained variance to R² = 0.174 (17,4%; must be >20%) (p = 0.008). In the final model, however, none of the individual predictors (age, BMI, SATAQ-3 subscales, PP, NP) reached statistical significance (all p > 0.05)'. Suggestion: No regression model seems to have helped the Authors to clarify the association between dependent and independent variables. Therefore, I suggest that the Authors should try to logarithmize the data (when using linear regression analysis) or perform logistic regression analysis as an alternative.
- Overall, the section of Results is very light; therefore, confusion should be reduced and initial empirical data reanalyzed, too.
- Line 324: 'Our findings align with this area, providing data on the prevalence of ON risk…'. The Authors should be aware that the concept 'prevalence' has to be changed to 'proportion'.
- Line 330: '…at risk of the disorder…'. The Authors should know that they write only about mental health symptoms. Disorders can only be diagnosed and confirmed by psychiatrists.
- Line 333: '…was significantly associated…' should be changed to '…correlated…'.
- Line 336: '…perfectionism was the most important block of predictors…'. The use of causal language must be declined.
- Line 347: '…In the context of gender difference…'. What gender differences are discussed? Perhaps the authors meant 'sex differences'.
- Line 355: … that risk factors manifest… What do the Authors think about these causal statements? It seems necessary to rephrase the entire Discussion section.
- Line 420: '…Moreover, the inclusion of objective body composition…'. The Authors did not assess the body composition. The Authors calculated only the body mass index.
- Line 433: '…Bivariate analyses suggested positive co-occurrence of orthorexic tendencies with perfectionism components,…'. The Authors should be aware that they have carried out a single cross-sectional study and therefore no changes can be described.
- Line 440: '…no single predictor maintained an independent association…'. What does this thesis, which the Authors presented in the conclusions section, mean?
- Line 450: '…allow for a more precise determination of the direction of associations and the mediating mechanisms…'. What does this thesis, which the Authors presented in the conclusions section, mean?
- Conclusions: The conclusions are unjustifiably and illogically globalized. The Authors should be aware that conclusions are not overwriting of results. The conclusions must address the hypotheses of this cross-sectional study. The conclusions must be overwritten by the Authors. Also, it is necessary to supplement the conclusions with further directions of research (e.g. in the case of cohort study in design). Specific practical recommendations need to be added to the conclusions. More specifically, there is a need to answer the question of what measures healthcare providers need to take in order to strengthen possible mental outcomes in a sample of adolescent football athletes.
- All in all, in the paper, there's still a lot of work to be done. The Authors should be confident that the Reviewer will re-review this manuscript. Therefore, all points must be responded and the manuscript should be revised according to all comments ans suggestions.
Author Response
Thank you so much for taking the time to evaluate our work. We have tried to incorporate all your valuable suggestions. If we could improve our work in any way, please let us know.
I have reviewed the manuscript titled 'Associations Between Sociocultural Attitudes Toward Appearance, Perfectionism, and Orthorexia Nervosa in Adolescent Football Athletes'.
As a consequence, I have serious concerns as follows:
- Suggestion: In the manuscript, the term 'orthorexia' could be changed to 'symptoms of orthorexia'.
Thank you for your comment. Corrected as suggested.
- I recommend that the Authors should return keywords in alphabetical order.
Thank you for your comment. Corrected as suggested.
- Lines 58-65: This information is more methodological oriented. I recommend that the Authors could move this paragraph to the section of Materials and Methods.
Thank you for your comment. Corrected as suggested.
- Lines 101-105: The Authors put forward a number of alternative hypotheses. Also, I recommend that the Authors could write down null hypotheses.
The description has been supplemented as suggested.
- 2.2. Participants: How was the representative sample size calculated? How large was the marginal error applied?
Thank you for your helpful comment. This study used a census of the finite target population within a single football academy rather than a draw from a larger regional/national frame. We invited all eligible players (N=85) and, after exclusions and absences, analyzed n=73 participants, inclusion rate: 85.9%. That is, representativeness is defined with respect to this academy’s roster, which we largely exhausted.
For proportions we report 95% confidence intervals using the finite population correction. The corresponding 95% margin of error (MOE) is:
MOE = 1.96 × sqrt[ p × (1 − p) / n ] × sqrt[ (N − n) / (N − 1) ]
where: p = proportion (0–1), n = sample size, N = population size.
In percent: MOE[%] = 1.96 × sqrt[ p × (1 − p) / n ] × sqrt[ (N − n) / (N − 1) ] × 100%
- 2.2. Participants: What was the technique used to select the study participants: probabilistic or non-probabilistic?
Participant selection was non-probabilistic, using a census (total population sampling) approach: all eligible players from a single academy were invited to participate; no random sampling was employed.
- 2.2. Participants: I propose that the number of participants (n) in the survey be adjusted in the light of the exclusion criteria.
The description has been amended in line with the reviewer's suggestion to make it more comprehensible to readers.
- Lines 150-153: Add the references, please.
Thank you for your comment. Corrected as suggested.
- Nutritional Status: The Authors should be aware that the body mass index is not a nutritional status. Additional research on the intake of macro-nutrients and eating habits appear necessary. Therefore, I suggest the Authors to decline writing about the nutritional status.
Thank you for this important point. We fully agree that BMI alone is not a comprehensive measure of nutritional status. At the same time, we would like to retain a brief description of BMI-based weight status because age- and sex-adjusted BMI-for-age is a well-established, widely used screening indicator in adolescent populations (as recommended/used by the World Health Organization), and in our study it serves only a descriptive/contextual role not a diagnostic assessment of nutrition. To avoid overinterpretation, we will revise the terminology and explicitly state its limitations. The term ‘nutritional status’ has been replaced with ‘BMI-based body weight status (screening indicator)’ throughout the text.
- Line 228: 'Associations between...' should be changed to 'Differences between...'.
Thank you for your comment. Corrected as suggested.
- Line 233: . '..predictors was assessed... ' should be changed '...potential predictors...'. The Authors should be aware that they have carried out a single cross-sectional study in design and that the use of the causal language must be minimized throughout the entire manuscript.
Thank you for your comment. Corrected as suggested.
- In the section, namely, Materials and Methods, must be indicated variables, which have been assigned to independent variables and dependent ones.
The description has been supplemented as suggested.
- Results: I recommend that all numbers (except p-value) should be rounded to one decimal place.
The description has been supplemented as suggested.
- Table 1: 'Age [years]' must changed to 'Age (years)'.
The description has been supplemented as suggested.
- Table 1: 'X—average'. This symbol 'X' is not the mean. This information needs to be corrected.
Thank you for the remark. We agree that “X” is not the conventional symbol for the sample mean. We have corrected Table 1 and the text: replaced “X” → “M (mean)”
- Lines 250-257: In the Reviewer’s view, the information on those lines is irrelevant.
Removed as suggested.
- Line 260: '…significant associations...'. The Authors should be aware that the identification of an association is a subject obtained from regression analysis. Otherwise, the concept 'correlation' should be used.
Thank you for your comment. Corrected as suggested.
- Line 267: 'The most frequently chosen platforms were Instagram (35.6%) and TikTok (34.2%), followed by Snapchat (11.0%)…'. Can the Authors explain why this information is important? I suggest that the Authors could write the Results section in a more consistent way and give only the most important results.
Thank you for your comment. The information about the most frequently used platforms was originally provided solely for contextual purposes – as background for variables related to the internalisation of socio-cultural norms (SATAQ-3), which operationalise exposure to content related to appearance/nutrition in the study.
- Line 279: 'Significant positive associations were observed…' must be changed to '…correlation…'.
Corrected as suggested.
- Line 305-307: 'Model 3 significantly increased the explained variance to R² = 0.174 (17,4%; must be >20%) (p = 0.008). In the final model, however, none of the individual predictors (age, BMI, SATAQ-3 subscales, PP, NP) reached statistical significance (all p > 0.05)'. Suggestion: No regression model seems to have helped the Authors to clarify the association between dependent and independent variables. Therefore, I suggest that the Authors should try to logarithmize the data (when using linear regression analysis) or perform logistic regression analysis as an alternative.
Thank you for the suggestion. We agree that low effect sizes should not be over-interpreted and that model assumptions should be verified. We therefore ran additional logistic regressions using the cut-offs DOS ≥25 (elevated risk) vs. <25 and DOS >30 (ON) vs. ≤30. The conclusions remain unchanged: no individual predictors were statistically significant, and the directions of effects were consistent with the linear model.
- Overall, the section of Results is very light; therefore, confusion should be reduced and initial empirical data reanalyzed, too.
Thank you for the comment. We intentionally kept the Results section concise to focus the reader on the key outcomes aligned with the study aim and to avoid repeating information already presented in the tables. We report all essential statistics (parameters, 95% CIs, p-values, R², and ΔR²), which minimizes ambiguity without introducing unnecessary digressions. For these reasons, we believe the current, succinct layout of the Results section accurately reflects the value and scope of the data obtained.
- Line 324: 'Our findings align with this area, providing data on the prevalence of ON risk…'. The Authors should be aware that the concept 'prevalence' has to be changed to 'proportion'.
Corrected as suggested.
- Line 330: '…at risk of the disorder…'. The Authors should know that they write only about mental health symptoms. Disorders can only be diagnosed and confirmed by psychiatrists.
Thank you for the clarification. We agree that our study concerns symptoms screened by questionnaires and does not support diagnostic statements. We have therefore replaced the phrase “at risk of the disorder” with “at elevated risk of orthorexic symptoms” throughout the manuscript.
- Line 333: '…was significantly associated…' should be changed to '…correlated…'.
Corrected as suggested.
- Line 336: '…perfectionism was the most important block of predictors…'. The use of causal language must be declined.
Thank you for the note. We have removed causal wording and rephrased the sentence to a neutral, statistical description.
- Line 347: '…In the context of gender difference…'. What gender differences are discussed? Perhaps the authors meant 'sex differences'.
Corrected as suggested.
- Line 355: … that risk factors manifest… What do the Authors think about these causal statements? It seems necessary to rephrase the entire Discussion section.
Thank you for pointing this out. We agree that, given our cross-sectional observational design, any causal language is inappropriate. We have therefore systematically rephrased the Discussion to describe associations rather than effects.
- Line 420: '…Moreover, the inclusion of objective body composition…'. The Authors did not assess the body composition. The Authors calculated only the body mass index.
We sincerely apologise for the typographical error, which has now been corrected.
- Line 433: '…Bivariate analyses suggested positive co-occurrence of orthorexic tendencies with perfectionism components,…'. The Authors should be aware that they have carried out a single cross-sectional study and therefore no changes can be described.
We removed any language implying change over time.
- Line 440: '…no single predictor maintained an independent association…'. What does this thesis, which the Authors presented in the conclusions section, mean?
We clarified that the block improved overall fit, but individual coefficients were not significant after mutual adjustment.
- Line 450: '…allow for a more precise determination of the direction of associations and the mediating mechanisms…'. What does this thesis, which the Authors presented in the conclusions section, mean?
We replaced the ambiguous phrasing with a non-causal, design-focused statement.
- Conclusions: The conclusions are unjustifiably and illogically globalized. The Authors should be aware that conclusions are not overwriting of results. The conclusions must address the hypotheses of this cross-sectional study. The conclusions must be overwritten by the Authors. Also, it is necessary to supplement the conclusions with further directions of research (e.g. in the case of cohort study in design). Specific practical recommendations need to be added to the conclusions. More specifically, there is a need to answer the question of what measures healthcare providers need to take in order to strengthen possible mental outcomes in a sample of adolescent football athletes.
Thank you for your suggestions. The applications have been rewritten to meet your expectations.
- All in all, in the paper, there's still a lot of work to be done. The Authors should be confident that the Reviewer will re-review this manuscript. Therefore, all points must be responded and the manuscript should be revised according to all comments ans suggestions.
Thank you for your insightful and very helpful review. All points have been taken into account and the manuscript has been comprehensively revised in accordance with your comments and suggestions. We enclose a detailed point-by-point response and a version of the manuscript with the changes marked to facilitate re-evaluation. Thank you for taking the time to evaluate our work and we remain at your disposal during the re-review.
Reviewer 2 Report
Comments and Suggestions for Authors
Associations Between Sociocultural Attitudes Toward Appearance, Perfectionism, and Orthorexia Nervosa in Adolescent Football Athletes
This study examined how Orthorexia Nervosa, sport-related perfectionism and internalized appearance attitudes are connected in 16–19-year-old football academy sample.
Introduction - In line with the study purpose. Relevant references.
Methodology -well detailed. Appropriate design. Limited sample that excludes the possibility of generalization of results. Define sampling method (strictly defined sampling procedure is not ok).
The objectives of the study are consistent with the results and conclusions
Unfortunately, other sociodemographic factors that could have influenced the results of the study (e.g. family support) are not taken into account – acknowledged in the limitations of study.
Conclusions – somehow overrated considering the sample size
Reviewer’s conclusions
Although some analysis show statistical significance, interpretation should be made with caution (findings are specific to this group …….)
Table 4 - all predictors should be translated
Author Response
Thank you so much for taking the time to evaluate our work. We have tried to incorporate all your valuable suggestions. If we could improve our work in any way, please let us know.
This study examined how Orthorexia Nervosa, sport-related perfectionism and internalized appearance attitudes are connected in 16–19-year-old football academy sample.
Introduction - In line with the study purpose. Relevant references.
Methodology -well detailed. Appropriate design. Limited sample that excludes the possibility of generalization of results. Define sampling method (strictly defined sampling procedure is not ok).
Thank you for the assessment. We agree that the limited, single-academy sample constrains external generalizability, and we have clarified the sampling method accordingly.
“Participant selection was non-probabilistic, using a census (total population sampling) approach: all eligible football players from a single Polish academy were invited (frame N = 85). After pre-specified exclusions (e.g., lack of consent, incomplete data), the final analytic sample was n = 73; throughout, n refers to this post-exclusion sample.”
We trust these changes address the reviewer’s concerns about design description and generalizability.
The objectives of the study are consistent with the results and conclusions
Unfortunately, other sociodemographic factors that could have influenced the results of the study (e.g. family support) are not taken into account – acknowledged in the limitations of study.
Conclusions – somehow overrated considering the sample size
Reviewer’s conclusions
Although some analysis show statistical significance, interpretation should be made with caution (findings are specific to this group …….)
Thank you for the insightful comments. We have thoroughly revised the Conclusions section in line with your suggestions: we reduced overgeneralization, emphasized the associational nature of the findings, clarified the limits of generalizability, and linked the conclusions directly to the study aims and tested hypotheses. We also expanded the Limitations to note unmeasured sociodemographic factors. Thank you for helping us improve the manuscript.
Table 4 - all predictors should be translated
Thank you for your comment. Corrected as suggested.
Reviewer 3 Report
Comments and Suggestions for Authors
Dear authors:
1 – The authors mention that, as detailed in the bibliography, the concept has "attracted considerable attention" and that "numerous studies indicate that orthorexia is associated with a pathological focus on the quality of food." However, readers need to examine the paper up to reference 15 to find a more recent reference from 2024. It is important to update the references as needed to accurately reflect how this issue is being discussed in the scientific community.
2 – The statistical values, such as p and r, should be in italic.
3- In the Discussion section, authors should include statistical values when comparing variables (for example, see line 327).
4- The reference at line 369 should follow the author’s name in line 367: Smith et al. [37]. Additional references in the text require similar corrections.
Best regards.
Author Response
Thank you so much for taking the time to evaluate our work. We have tried to incorporate all your valuable suggestions. If we could improve our work in any way, please let us know.
1 – The authors mention that, as detailed in the bibliography, the concept has "attracted considerable attention" and that "numerous studies indicate that orthorexia is associated with a pathological focus on the quality of food." However, readers need to examine the paper up to reference 15 to find a more recent reference from 2024. It is important to update the references as needed to accurately reflect how this issue is being discussed in the scientific community.
Thank you for this remark. We have updated the reference list to include the most recent sources from 2020–2025. Older references were retained only where historically necessary; elsewhere they have been replaced or supplemented with up-to-date reviews and empirical studies.
2 – The statistical values, such as p and r, should be in italic.
Thank you for your comment. Corrected as suggested.
3- In the Discussion section, authors should include statistical values when comparing variables (for example, see line 327).
Thank you for your comment. Corrected as suggested.
4- The reference at line 369 should follow the author’s name in line 367: Smith et al. [37]. Additional references in the text require similar corrections.
Thank you for your comment. Corrected as suggested.
Round 2
Reviewer 1 Report
Comments and Suggestions for Authors
The Authors improved the paper.